# Tannic Acid and Tea Prevents the Accumulation of Lead and Cadmium in the Lungs, Heart and Brain of Adolescent Male Wistar Rats—Possible Therapeutic Option

**DOI:** 10.3390/ani12202838

**Published:** 2022-10-19

**Authors:** Anna Winiarska-Mieczan, Małgorzata Kwiecień, Maciej Bąkowski, Robert Krusiński, Karolina Jachimowicz-Rogowska, Marta Demkowska-Kutrzepa, Bożena Kiczorowska, Wanda Krupa

**Affiliations:** 1Institute of Animal Nutrition and Bromatology, University of Life Sciences in Lublin, Akademicka 13, 20-950 Lublin, Poland; 2Department of Parasitology and Invasive Diseases, Faculty of Veterinary Medicine University of Life Sciences in Lublin, Akademicka 13, 20-950 Lublin, Poland; 3Department of Animal Ethology and Wildlife Management, University of Life Sciences in Lublin, Akademicka 12, 20-950 Lublin, Poland

**Keywords:** tannic acid, tea, antioxidant activity, lead, cadmium, rats organs, adolescent rats

## Abstract

**Simple Summary:**

Polyphenols, including tannic acid, have strong antioxidant properties. They chelate prooxidant toxic metals and are known as pro-oxidant/antioxidant balance agents. Tea aids detoxification of the body by inhibiting absorption and facilitating excretion of toxic metals from the body. Since the tea infusions acted more effectively than the isolated TA, it can be assumed that the protective effect of teas on the organism against toxic metals should be considered in the context of the cumulative effect of various active substances present in the infusions.

**Abstract:**

The protective effect of tannic acid and tea solutions on the lungs, hearts and brains of adolescent Wistar rats exposed to Pb and Cd was studied. Metals were administered with feed (7 mg Cd and 50 mg Pb/kg). Two experiments were carried. Experiment 1 aimed to determine the level of tannic acid (TA), most effectively reducing the adverse impact of Pb and Cd on the organs of adolescent rats (aged 5 weeks, weighing 169.3 ± 14.7 g) during combined exposure. TA was administered with drink (0, 0.5, 1, 1.5, 2 or 2.5% solutions). In Experiment 2, adolescent rats (aged 6 weeks, weighing 210.6 ± 12.1 g) received an aqueous solutions of black, green, red or white teas. TA and teas had a positive effect on reducing the accumulation of Cd in the organs. The results obtained suggest that long-term continuing administration of TA increases its effectiveness as a chelator for Pb. A 2% TA and white tea solution proved to be the most effective. In the analyzed tissues, increased activity of SOD and CAT was recorded as a result of the use of the TA and teas; thus, they can efficiently prevent the prooxidant effect of toxic metals.

## 1. Introduction

The presence of toxic metals in food products has become a global problem. The most important source of toxic metals for man is food of plant origin, and in particular cereals [1], mostly with regard to the fact that they are the basis of nourishment throughout the world and are consumed most abundantly. The half-life of Cd in soft tissues is between 25–30 years [2], while that of Pb is about 30 days [3]. In 2012, the European Food Safety Authority (EFSA) [1,4] reduced the tolerable intake level for Cd and Pb. The dietary Cd tolerable weekly intake (TWI) was estimated at 2.5 μg/kg of body weight/week [4], whereas the benchmark dose lower confidence limit (BMDL) for Pb was: BMDL_01_ 1.5 μg and BMDL_10_ 0.63 μg/kg of body weight/day in adults, and BMDL_10_ 0.5 μg/kg of body weight/day in children [1]. The environmental exposure of humans to cadmium is approximately 5 mg/kg of body weight/week, and to lead, it is approximately 35 mg/kg of body weight/week [5]. Studies in rats have shown substantially higher accumulation of Cd and Pb in organs when these metals are administered in water than in food [6]. The distribution of Pb and Cd, depends primarily on their ability to penetrate biological membranes and to some extent on the affinity with the specific structure of the organism [7]. Resorption depends on the form of Pb and Cd, metabolic activity of the organism, age and physiology [8,9,10,11].

Pb and Cd can cause both morphological lesions and disorders of cardiac muscle (contractility, conduction), lungs and brain damage as a result of oxidative stress [12,13]. Cd and Pb, despite not actively participating in redox reactions, are indirectly conducive to oxidative stress, generated, among other things, by increasing the peroxidation of lipids and inhibiting the activity of antioxidant proteins through binding them with sulfhydryl groups and disturbing calcium homeostasis [14]. Under the influence of heavy metals, the concentration of reactive forms of oxygen, which can result from the release of transitory metals from places in which they naturally occur within a cell increases [15,16]. The study by Jedlińska-Krakowska [17] revealed that the hearts of rats receiving Cd, Pb and simultaneously Cd and Pb in the form of water drinking solutions had higher concentrations of malondialdehyde, which is one of the end products of polyunsaturated fatty acid peroxidation in cells and thus a marker of oxidative stress in humans and animals. In rats poisoned with Cd and Pb, reduced activity of antioxidant enzymes was found [18], which can show that antioxidant mechanisms become exhausted as the poisoning progresses. Cytotoxicity of xenobiotics is limited by antioxidant enzymes such as SOD (superoxide dismutase) and CAT (catalase) [14]. The use of exogenous antioxidants increases the total antioxidant capacity of the body and reduces the likelihood of inducing oxidative damage [19,20].

Since it is impossible to eliminate Pb and Cd from food, research is carried out on methods of limiting its absorption by the body. These methods should be easily accessible and uncomplicated [21,22,23,24,25,26,27,28,29]. Foods containing significant amounts of antioxidant components can be used in the daily diet to prevent the dangerous effects of toxic metals on the human body [30,31,32,33,34,35,36,37,38,39,40]. Tea deserves particular attention. The most frequently drunk varieties are black and green tea [41,42]. Tea satisfies the conditions for classification as a functional food. In addition, it was found that tea infusions enhance detoxification processes in rats by binding toxic metals and intensifying their excretion from the body [15]. Tea contains a number of substances which are capable of chelating metals, such as, tannic acid (TA) [43], catechins, quercetin and others polyphenols [44]. Polyphenols have strong antioxidant properties [45], they chelate prooxidant metals and they are known as the agents of the pro-oxidant/antioxidant balance [15,46]. Moreover, studies have demonstrated that 60 min after drinking green tea, human blood serum contains three times more catechins than after drinking black tea [47]; thus, the antioxidant potential in blood serum after drinking green tea is increased by 34%, while after drinking black tea, the increase is 29% [48]. Drinking green tea in an amount of four brewing sachets per day for 12 weeks increased antioxidative capacity of saliva by 42% [49]. Daily intake of 3 to 5 cups of green tea could provide about 250 mg of catechins [50].

Extensive research was carried out investigating the impact of Pb and Cd both in people and in animals, which enabled approximate determination of toxic activity mechanisms. However, this research has mostly involved a model of poisoning with large doses of metals, which accelerates negative effects that, in terms of their intensity, reveal a whole range of responses of the body both in terms of destruction and defense mechanisms. However, the results of this research do not illustrate the actual interaction in the human environment. It should be remembered that people are mostly exposed to these factors occurring in low concentrations and their effects are revealed only when irreversible impairment occurs. It seems extremely significant to know the response of the body to toxins (including toxic metals) at small doses, which would enable effective protection against further adverse consequences or increase the effectiveness of therapies used.

Tannins are more effective antioxidants than, for example, vitamin C and E [51]. Tannic acid (TA) in in vitro conditions is a very efficient chelator for Pb [52]. It is known from our previous studies that a 2% TA drinking solution given to adult rats exposed to Pb and Cd in an amount not exceeding 18.5 mg Pb and 2.6 mg Cd/kg body weight/week is an effective way to reduce Cd accumulation in the brain; however, TA may not be as effective for Pb [53]. This study aimed to verify whether TA and tea administered to rats orally in the form of a water solution will have any influence on the adverse effect of Cd and Pb in adolescent rats chronically exposed to Pb and Cd in doses not exceeding the levels of environmental exposure of humans. Using a rat model, this study aimed to: (1) determine the level of TA, most effectively reducing the adverse impact of Pb and Cd on the lungs, hearts and brains of adolescent rats during combined exposure: the degree of reduction of the accumulation of Pb and Cd in the analyzed tissues and the activity of antioxidant enzymes (SOD and CAT) and (2) check whether administering of black, green, red and white tea solutions containing the specific level of TA would decrease the absorption of Pb and Cd and increase the activity of SOD and CAT in organs; this study assumed that TA is one of the crucial components of tea determining the preventive effect of tea in relation to Pb and Cd.

## 2. Materials and Methods

Two experiments were carried. The experiments were approved by the II Ethical Committee at the University of Life Sciences in Lublin (No. 18/2010). All animal procedures conform to the principles embodied in the Declaration of Helsinki and all procedures related to animal use comply with the Guiding Principles for Biomedical Research Involving Animals. The proper experiments took 12 weeks: the first period covering weeks 1–6 and the second period covering weeks 7–12 (Figure 1). The rats had unlimited access to feed and drinking solutions. After one week of adaptation, the animals were housed separately in propylene cages in rooms under standard environmental conditions: temperature of ca. 21 °C, humidity of ca. 55%. A 12 h light/dark cycle was maintained. In the experiment, Cd and Pb doses not exceeding the environmental exposure limits for humans were adopted, since in real life such a situation occurs more commonly than poisoning with toxic doses. There is little information on the effects of exposure to low doses of toxic metals. Throughout the term of the experiment, the rats were fed with a mix produced at our laboratory: standard feed for laboratory animals was ground and mixed with 50 mg Pb/kg as (CH_3_COO)_2_Pb and 7 mg Cd/kg as CdCl_2_ by mechanical methods and granulated. The consumption of feed and drink was recorded every 7 days. Based on the weekly consumption of drinking fluids, the intake of TA was calculated. For the calculation of Pb and Cd intake, the mean weekly food consumption was used. The rats were weighed every 7 days. In each group 6 rats, both after the 6th and after the 12th week of the experiment, were weighed, and next they were sacrificed. The rats were put down in CO_2_ and they were sacrificed by their spinal cord being broken. Immediately after euthanasia and bloodying of the rats their brains, lungs and hearts were prepared as a whole, weighed and washed using chilled saline solution, and placed in plastic containers and frozen at—20 °C for chemical analysis.

### 2.1. Animals and Treatments

#### 2.1.1. Experiment 1

Experiment 1 aimed to determine the level of TA, most effectively reducing the adverse impact of Pb and Cd on the lungs, hearts and brains of adolescent rats during combined exposure: the degree of reduction of the accumulation of Pb and Cd in the analyzed tissues (for 6 and 12 weeks of exposure) and the activity of antioxidant enzymes (SOD and CAT), compared with the Control group. 72 adolescents male Wistar rats (aged 5 weeks, weighing 169.3 ± 14.7 g) were used for this study. The animals were divided into six groups (one control and five experimental groups), each of 12 rats, as follows: Control group receiving distilled water; Group 0.5% TA receiving 0.5% aqueous solution of TA; Group 1% TA receiving 1% aqueous solution of TA; Group 1.5% TA receiving 1.5% aqueous solution of TA; Group 2% TA receiving 2% aqueous solution of TA; Group 2.5% TA receiving 2.5% aqueous solution of TA.

#### 2.1.2. Experiment 2

This study assumed that TA is one of the crucial components of tea determining the preventive effect of tea in relation to Cd and Pb. Adolescent male Wistar rats (aged 6 weeks, weighing 210.6 ± 12.1 g) were administered black, green, red and white tea. Cd and Pb levels (after 6 and 12 weeks of exposure) as well as SOD and CAT activity in the lungs, hearts and brains during the combined exposure were examined in comparison with the Control group. The rats were randomly assigned to 5 equivalent groups (of 12 rats each): Control—drank distilled water (no treatment); BT—drank black tea (Yellow Label); GT—drank green tea; RT—drank red tea (Pu-erh); WT—drank white tea. Since Experiment 1 showed that the optimum level of TA was 2%, and that rats were not willing to drink solutions containing more than 2% of TA; the conclusion is that the content of TA in experimental tea solutions should be 2%. The tea infusions were made by soaking tea bags purchased from a commercial source in distilled water at 90 °C for 5 min, retaining the proportion of 1 tea bag per 200 mL of water. In the resultant infusions, the content of TA was determined by the spectrophotometric method. Tea infusion of 100 mL contained: 94 ± 3.4 mg of TA in black tea (India, Lipton), 111 ± 1.8 mg of TA in green tea (China, Lipton), 77 ± 1.2 mg of TA in red tea (China, Lipton), and 113 ± 2.3 mg of TA in white tea (China, Lipton). Total polyphenol content (as TA equivalent) in 100 mL of the tea infusions was: 122 mg in black, 263.3 mg in green, 99.6 mg in red and 266.8 mg in white. The infusions were divided into 30 mL samples which were frozen at −20 °C. The frozen samples were used every time to prepare solutions ensuring that the drinking solutions used throughout the term of the experiment had identical TA concentration.

### 2.2. Chemical Analyses

#### 2.2.1. Determination of the Content of Pb and Cd in the Lungs, Brains and Hearts

The tissue samples were weighed (approximately 3 g each), and subjected to mineralization. The samples were dried at 105 °C for 48 h to constant weight and dry-mineralized in a muffle furnace at 450 °C for 12 h, with hydrogen peroxide as the oxidant. Ash samples were dissolved in 10 mL of 1 M HNO3. A Varian SpectrAA 880 atomic absorption spectrophotometer was used for the determination of Pb and Cd (Table 1). The reliability of the analytical measurements was checked by analyzing certified reference material (CRM-185R—bovine liver contained Cd 0.544 mg/kg and Pb 0.172 mg/kg) and blank samples. Calibration lines were obtained using the standard solutions of Pb and Cd (Merck, Germany) in the concentrations: 0.00, 0.10, 0.20, 0.40, 1.00 and 2.00 ng/mL. Each sample was analyzed in three replications.

#### 2.2.2. Determination of Antioxidant Enzymes Activity

The activity of SOD and CAT in the lung, heart and brain homogenates was determined solely after 12 weeks of the experiment. Tissue samples were homogenized in a 0.9% NaCl solution at 4 °C. The homogenates were centrifuged at 4500× *g* for 20 min (4 °C). In the resulting supernatants, the activity of SOD was determined by the adrenaline method of Misra and Fridovich [54], and absorbance was measured using a spectrophotometer at λ = 480 nm. CAT activity in the supernatants was determined using the method described by Sinha [55], by means of a spectrophotometer at wave length λ = 570 nm. Quality control of analytical measurements was performed using blank samples and a standard substrate (hydrogen peroxide) at the concentrations of 20, 40, 60, 80 and 100 µmoles.

#### 2.2.3. Determination of the Content of TA in Tea Infusions

The content of TA was determined by the spectrophotometric method. TA was extracted by the mixture of equal proportions of ethyl alcohol, glycerin and distilled water, and a colored TA complex was created with the Folin–Ciocalteu reagent. Extinction was measured using a spectrocolorimeter with EK-5 attachment, with 5 cm trays, at λ = 700 nm. Calibration lines were obtained using the standard solutions of TA (POCH, Poland) in the concentrations: 0, 10, 20, 30, 40 and 50 µg/mL. Quality control was performed using blank samples. Each sample was analyzed in three replications.

#### 2.2.4. Chemical Reagents

Tannic acid (C_76_H_52_O_46_), CdCl_2_ and (CH_3_COO)_2_Pb were purchased from POCH S.A. (Poland).

### 2.3. Calculations and Statistical Analysis

Lungs, heart and brain relative weights (%) are expressed as organ weight × 100/final body weight. Statistical values were expressed as the mean value ± standard deviation (SD). The content of Pb and Cd in the tissues was expressed as μg/g of wet tissue. The effectiveness of TA solutions was evaluated according to the degree of reduction in absorbing Pb and Cd in the tissues compared with the appropriate Control groups, where the values were assumed as 100%. The data was statistically analyzed using STATISTICA 6.0 software. Statistically significant differences were established at the level of *p* < 0.05. The significance of differences between the mean values in the groups was estimated with the one-way ANOVA using the *t*-Student’s Newman–Keuls test.

## 3. Results

### 3.1. Experiment 1

#### 3.1.1. Drinking Fluids and Feed Consumption and Body Weight Gain

Based on the amount of water solutions consumed (Table 2), it can be observed that rats were not willing to drink the solution containing 2.5% TA, which in turn contributed to significantly reduced consumption of TA by rats in this group compared to other groups. The feed intake was also significantly lower in this group than in other experimental groups, which resulted in lower body weight gain (Table 3).

#### 3.1.2. The Relative Weight of Organs

In both experiment periods, the relative weight of the analyzed organs was even in experimental groups. A significantly higher value was recorded only in rats from the 2.5% TA group (Table 4).

#### 3.1.3. The Distribution of Pb and Cd in Rats’ Lungs, Hearts and Brains

The highest accumulation of Pb and Cd was found in the brains, followed by the lungs and the hearts. Figure 2 presents the content of Pb and Cd in the analyzed rat tissues. In both experimental periods, no statistically significant effects of TA on the content of Pb in the lungs of rats were identified. The brains of rats from the 2% TA and 2.5% TA group and the hearts from 2.5 TA group in the second experimental period revealed a significantly reduced content of Pb compared to the Control group. The level of Cd, compared to that in the Control group in both experimental periods, was lower (*p* < 0.05) than in all the analyzed organs of rats from the 1.5% TA, 2% TA and 2.5% TA group. In addition, a significantly lower level of Cd compared to that in the Control group was recorded in the hearts of rats from the 1% TA group in the second experimental period. This study evaluated the effectiveness of TA at different concentrations based on the degree of reduction (%) in the level of accumulation of Pb and Cd in the lungs, hearts and brains, compared with the Control group (Table 5). Only in the brains of rats from the 2% TA and 2.5% TA and in the hearts of rats from the 2.5% TA groups after 12 weeks, a less than 6% reduction in the concentration of Pb in comparison with the Control group was recorded. In both periods of the experiment, the highest effectiveness of reducing the absorption of Cd was revealed in the organs of rats from 2% TA and 2.5% TA, approximately 12—22% in the first period and 19–32% in the second period. The TA solutions were the most effective in the case of lungs.

#### 3.1.4. Activity of Antioxidant Enzymes in the Lungs, Hearts and Brains

In the analyzed tissues, increased activity of both SOD and CAT was recorded as the result of the use of TA (Figure 3). In the lungs, a statistically significant increase in the activity of SOD, compared to the Control group, was observed in rats from the 2% TA and 2.5% TA groups, while in the hearts and brains, the increase (*p* < 0.05) was noted in the 1.5% TA, 2% TA and 2.5% TA groups. Increased (*p* < 0.05) CAT activity was observed in the organs of rats from the 1.5% TA, 2% TA and 2.5% TA group and, in addition, in the hearts of rats from the 1% TA group.

#### 3.1.5. The Effect of TA on the Lungs, Heart and Brain of Adolescent Rats—Conclusions from Experiment 1

In both experimental periods the use of a 2% and 2.5% TA solutions proved to be the most effective way of reducing the accumulation of Cd in the lungs, hearts and brains of adolescent rats compared with rats from the Control group (*p* < 0.05). No statistically significant effects of TA on the content of Pb in the organs of rats were identified, excluding the brains from the 2% TA and 2.5% TA groups, and hearts from the 2.5% TA group in the second period of the experiment. At the same time, it can be observed that rats were not willing to drink the solution containing 2.5% TA; the feed intake was also significantly lower in this group than in other experimental groups, which resulted in lower body weight gain. Therefore, the best results (concentration of Cd and Pb, activity of SOD and CAT, brain to body weight ratio) were obtained in the 2% TA group, regardless of the experimental period (weeks 1–6 or 7–12). Taking into account both these results and the claims of some authors [48,49] that TA is the major factor behind the bitter taste of beverages, it was determined that the content of TA in tea solutions in Experiment 2 should be 2%.

### 3.2. Experiment 2

#### 3.2.1. Drinking Fluids and Feed Consumption and Body Weight Gain

In both experimental periods, the highest intake of tea was recorded in the RT and WT groups (Table 2). In addition, after 12 weeks of the experiment no significant differences in the intake of the solution were noted between the GT and the Control. In the first experimental period (weeks 1–6), the feed intake was significantly higher in the WT group, whereas in the second period (weeks 7–12) the rats from all experimental groups consumed on average significantly (*p* < 0.05) less feed in comparison to the Control group. After 12 weeks of exposure, the heaviest body weight was found in rats from the GT group (Table 3).

#### 3.2.2. The Relative Weight of Organs

No differences (*p* < 0.05) in the mean relative weights of lungs, hearts and brains were observed between the experimental groups in both experiment periods as shown in Table 6.

#### 3.2.3. The Distribution of Pb and Cd in Rats’ Lungs, Hearts and Brains

Levels of both Pb and Cd in organs usually follow the ranking: brains, lungs, hearts. In the first period of exposure (weeks 1–6), the effect of tea solutions on the accumulation of Pb was not statistically significant, whereas after 12 weeks in the hearts and brains of rats from the WT group, a decreased (*p* < 0.05) level of Pb was recorded compared to the Control (Figure 4). A significantly lower level of Cd compared to that in the Control was observed in the organs of rats from all the experimental groups in both experimental periods. The effectiveness of tea solutions was evaluated based on the degree of reduction in the accumulation of Cd and Pb in the organs compared to the Control (Table 7). The effectiveness of tea solutions in relation to Pb was lower than in relation to Cd. Reduced absorption of Cd was recorded for all the examined organs. After 6 weeks of exposure, an approximately 15–22% lower absorption of Cd was observed in all experimental groups, while after 12 weeks these values were 17–29%. In both periods of exposure, the absorption of Cd in the organs in the WT group significantly improved compared to the remaining experimental groups. The tea solutions were the most effective in the case of lungs.

#### 3.2.4. Content of Antioxidant Enzymes in the Lungs, Hearts and Brains

In rats from the GT, RT and WT groups, SOD activity was increased in the examined organs. In addition, the activity of this enzyme was increased in the lungs and brains of rats from the BT group (Figure 5a). CAT activity in the lungs, brains and hearts of rats from all experimental groups was significantly increased compared to the Control (Figure 5b).

## 4. Discussion

In this study, the effectiveness of TA and teas was evaluated based on the percent of reduction in the accumulation of Cd and Pb in lungs, hearts and brains, compared with the Control group, as a result of the chelating properties of polyphenols. It should be taken into account that Experiment 1 (TA) and Experiment 2 (tea) in the present study were not conducted at the same time, although both rats and the experimental environment were similar in both experiments. However, since the effectiveness of TA solutions and teas was assessed based on the degree of reduction (%) of the Cd and Pb content in the organs compared to the control group (receiving distilled water to drink), where the value was assumed to be 100%, the results of the effectiveness in both experiments were determined as comparable.

There is little information in the available literature on the effect of TA and tea infusions on the accumulation of Cd and Pb in the tissues of laboratory animals. Most studies refer to green tea which supports the detoxification of the body by inhibiting the absorption of toxic metals and facilitating their excretion from the body [15]. This is due to the chelating action of polyphenols, mainly catechins, including EGCG that is abundant in green tea, quercetin and TA [15,52,56,57,58,59]. This study showed the highest effectiveness of reducing the absorption of Cd and Pb in the case of white and green tea solution, while black tea was the least effective. This should be attributed to the fact that white and green tea (non-fermented) contains more polyphenols, including catechins, capable of chelating metallic elements, than in black (fermented) and red (partially fermented) tea [60]. However, the protective effect of teas on the body against toxic metals must be considered in the context of the summative effect of various active substances present in infusions, as their effect is summative. Since non-fermented white and green tea is considerably richer in antioxidant substances than other teas, the most favorable results achieved in our presented studies are not surprising. For this reason, it can be assumed that the better results in our own study were obtained using tea solutions rather than isolated TA. Although, the concentration of TA in the experimental solutions was much higher than in the teas: 5000 mg/L in the 0.5%TA group—25,000 mg/L in the 2.5%TA group compared to 770–1130 mg/L in the groups receiving tea infusions.

Catechins, believed to be the most important antioxidants in tea, and in particular green tea, were thoroughly studied. However, the very strong antioxidant properties of TA is completely underestimated. TA is characterized by a higher antioxidant capacity than other polyphenols and this capacity is not lower than that of BHA, BHT and α-tocopherol [61]. TA in in vitro conditions inhibits the peroxidation of lipids almost 98% at the concentration of 15 μg/mL, while standard antioxidants (e.g., BHA and α-tocopherol) have similar results at the concentration of 45 μg/mL. TA also demonstrates a comparable capability of chelating metals [61]. TA, degraded in the gut by bacteria and enzymes, is absorbed from the alimentary tract in animals, it is found in blood plasma [62] and can have a chelating effect on the toxic metals present in blood and internal organs [57]. Thus, TA is apparently one of the crucial components of tea determining the preventive effect of tea in relation to Cd and Pb. Studies on rats showed that TA has an effect preventing the absorption of Cd and Pb [34] by binding these toxic metals and thus inhibiting their absorption into tissues. Kim et al. [63] demonstrated that the tissues of mice poisoned with Cd and given TA contained less Cd. They administered Cd (20 mg/kg) and/or TA (0.5 mg/mL, 1 mg/mL, or 2 mg/mL) water per os to mice for 4 weeks. In turn, Peaslee and Einhellig [56] showed that TA effectively inhibited Pb absorption in mice fed a diet containing both TA and Pb. Pekdemir et al. [52] demonstrated that TA was a very effective chelator of Cd and Pb in vitro. Similarly, our previous studies showed the effectiveness of TA and/or tea infusions in reducing the accumulation of Cd and/or Pb in the following rat tissues: lungs, brain, heart, liver, kidneys, blood and bones [15,64,65,66,67].

In these studies, TA and tea efficiently reduced the degree of accumulation of Cd in tissues, and Pb was less susceptible. Another study on rats showed that Pb was a metal more resistant than Cd to being bound by TA [30], where the use of TA solutions with the concentration from 0.5% to 2% in adult rats simultaneously exposed to Cd and Pb resulted in a statistically significant reduction of Cd absorption in brains but had no effect on the level of Pb. This could be due to the fact that Pb reveals a considerably stronger affinity with thiol groups [62], which do not occur in polyphenolic compounds, than with hydroxyl groups present therein. It is noticeable that the effectiveness of TA and teas in relation to Cd and Pb was higher in the second experimental period than in the first one. The results obtained suggest that long-term continuing administration of polyphenols increases its effectiveness as a chelator for Cd and Pb. Moreover, it cannot be excluded that if the term of the experiment was longer (>12 weeks), the effects for Pb would be more explicit. Further investigation is required to clarify this problem, since available results obtained by other authors, who unfortunately are scarce, indicate that Pb is a metal poorly susceptible to the chelating effect of orally administered antioxidants.

Teas were the least effective in the case of lungs compared to other organs. The results referred to both Cd and Pb. This can mean that drinking tea will not considerably change, for example, the adverse impact of smoking tobacco that is one of the most important sources of Cd and Pb for humans and simultaneously the most important cause of lungs cancer [68]. The polyphenols present in tea reveal strong anticarcinogenic properties [69,70], which in this case may not be fully utilized.

The system of endogenous antioxidant enzymes is supported by exogenous antioxidants. Supposedly, the decreased likelihood of a reaction between metals and critical biomolecules and inducing of oxidative damage is connected with the effect of antioxidant compounds contained in food or supplements [15,71]. The authors’ own research has demonstrated that the activity of SOD and CAT in the lungs, hearts and brains of the examined rats increased, which can provide evidence that TA and other polyphenols reduces the toxic effect of Pb and Cd on the activity of these metals. The highest increase in activity was recorded in rats receiving a 2% TA solution and green tea infusion. Polyphenols, including TA, have strong antioxidant properties [61,72]; they chelate prooxidant metals and they are known as the agents of the pro-oxidant/antioxidant balance [15]. Research by Guliasar et al. [73] demonstrated that the use of antioxidant substances (such as melatonin and taurine) significantly reduced the concentration of Cd in the brains and lungs of rats receiving a water drinking solution containing 200 μg/mL of Cd over 3 months. Significantly, the results for the heart were more favorable than for the brain, which is particularly important since the studies by the above-named authors revealed that the accumulation of Cd in hearts was several times higher than in brains. In turn, the authors’ own research shows that the organ most susceptible to the detoxicating effect of TA was the brain, where the accumulation of Cd was higher than in other tissues. The results of the authors’ own research and of the study by Guliasar et al. [73] can evidence that substances chelating toxic metals, when administered orally, are more effective for higher concentrations of these metals in the tissues. In addition, based on the results of our previous studies, it can be supposed that drinks containing TA, for example tea, more effectively reduce the accumulation of Cd in tissues when drunk along with consuming food contaminated with Cd than when drunk in between meals [34].

Exposing rats to Cd and Pb reduces the activity of antioxidant enzymes, which points to a decrease in the antioxidant potential of the body as a result of supplying factors enhancing cellular oxidation [15,71]. The present study showed an increase in the SOD and CAT activity in the lungs, brain and hearts of rats receiving tea, which was probably related to the enhancement of the antioxidant capacity of the organism through the supply of exogenous antioxidants. Our earlier research demonstrated an increase in the level of endogenous antioxidants only upon long-term exposure [65,66], which indicates that positive effects can only be achieved through the regular consumption of tea. El- Sayed et al. [46] revealed that green tea extract administered to rats receiving Cd in the form of a water-based solution containing 0.4% CdCl_2_ reduced the degree of peroxidation of lipids in testicles, thus preventing damage. According to Khalaf et al. [74], green tea extract administered to rats poisoned with Pb in the amount of 100 mg/kg of body weight over 15 days increased the activity of antioxidant enzymes, including SOD, in the brain. Wei and Meng [75] and Areba et al. [76] revealed that in rats poisoned with Pb, the activity of antioxidant factors after the use of EGCG was increased.

## 5. Conclusions

Both TA and tea reduced the negative effects of Cd and Pb on the rats’ bodies. Better effects were obtained during the longer duration of the experiment. This can be considered a promising therapeutic option in the case of environmental exposure to Cd and Pb. Since tea infusions acted more effectively than isolated TA, it can be assumed that the protective effect of teas on the body against toxic metals should be considered in the context of the summative effect of various active substances present in infusions. It would be interesting to identify the most effective ligand for Cd and Pb among tea components: TA, catechins and quercetin. The effectiveness of the most important components of tea, served separately, on the reduced accumulation of Cd and Pb in rats’ tissues should be compared.

## Figures and Tables

**Figure 1 animals-12-02838-f001:**
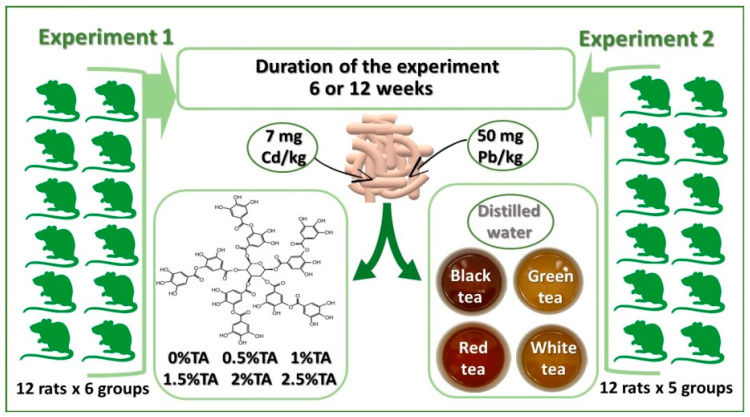
Experimental design. TA—tannic acid.

**Figure 2 animals-12-02838-f002:**
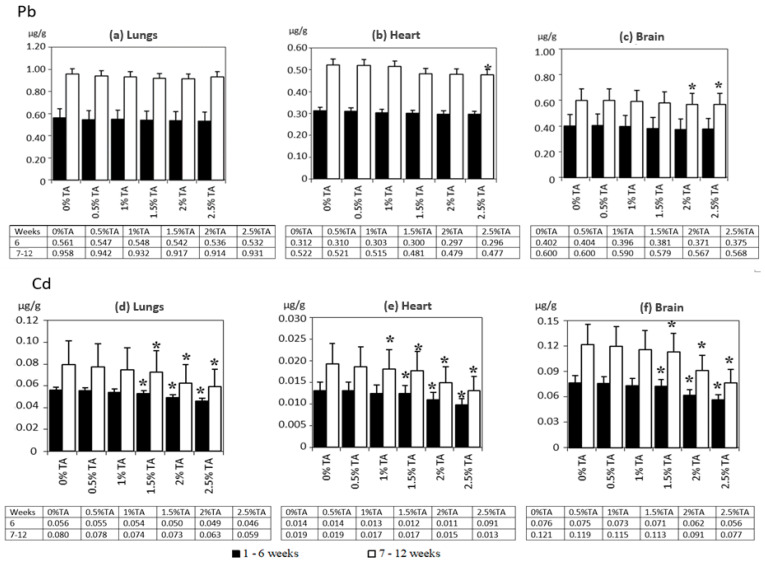
Experiment 1—The distribution of Pb and Cd in the lungs, heart and brain for 6 and 12 weeks of exposure (mean ± SD). SD—standard deviation; * significant (*p* < 0.05) versus 0%TA group.

**Figure 3 animals-12-02838-f003:**
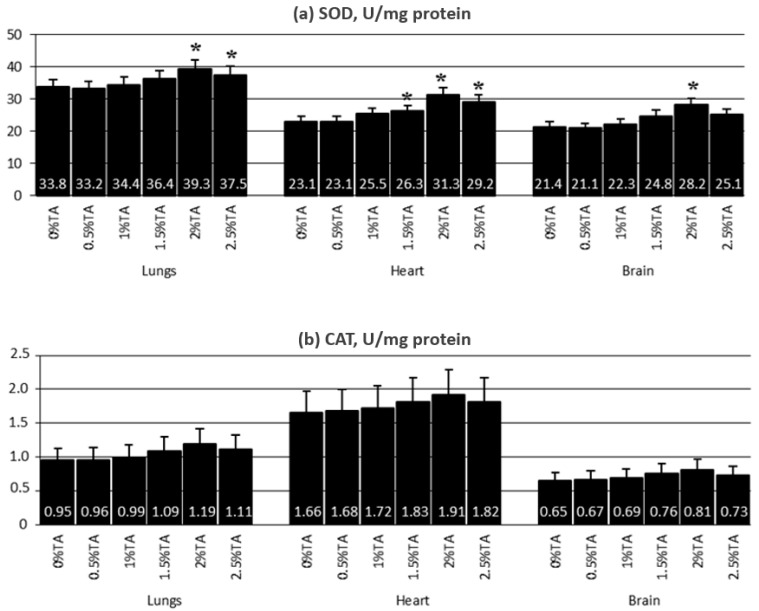
Experiment 1—Mean values of SOD and CAT activity in the lungs, heart and brain for 12 weeks of exposure (mean ± SD), U/mg protein. SD—standard deviation; * significant versus 0% TA (*p* < 0.05).

**Figure 4 animals-12-02838-f004:**
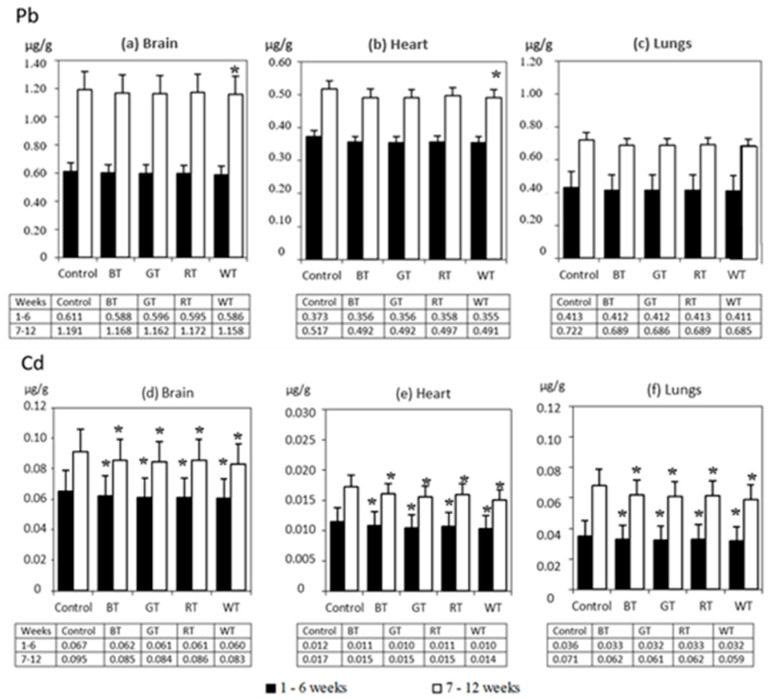
Experiment 2—The distribution of Pb and Cd in the lungs, heart and brain for 6 and 12 weeks of exposure (mean ± SD). SD—standard deviation; * significant versus 0% TA (*p* < 0.05).

**Figure 5 animals-12-02838-f005:**
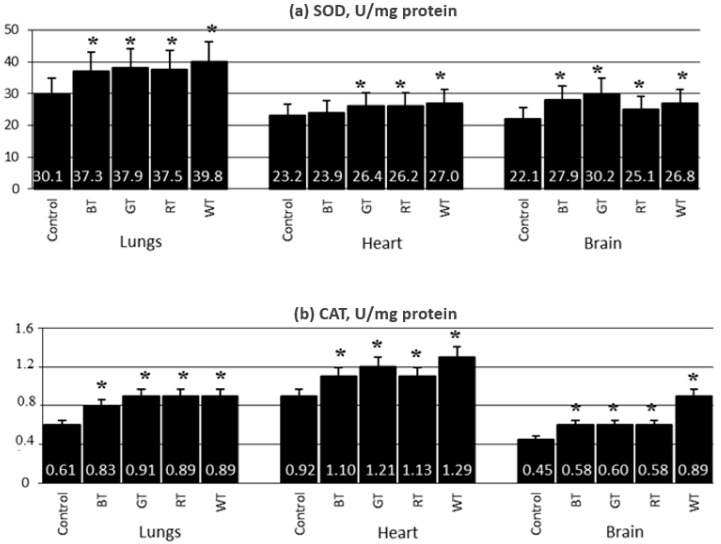
Experiment 2—Mean values of SOD and CAT activity in the lungs, heart and brain for 12 weeks of exposure (mean ± SD). SD—standard deviation; * significant versus 0% TA (*p* < 0.05), BT—black tea treated group, GT—green tea treated group; RT—red tea treated group; WT—white tea treated group.

**Table 1 animals-12-02838-t001:** Measurement parameters for the determination of Pb and Cd.

	Pb	Cd
Wave length, nm	217.0	228.8
Lamp current, mA	10	4
Spectral band pass, nm	1	0.5
LOD, mg/kg	0.011	0.001
LOQ, mg/kg	0.03	0.004
Pure gas	Argon	Argon
Background correction	Zeeman	Zeeman
Mean recovery rate	95%	96%
The deviation of duplicate measurement	5.2%	5.0%

**Table 2 animals-12-02838-t002:** Mean weekly feed, drinking fluids and TA consumption, total intake of Pb and Cd for 6 and 12 weeks of exposure.

Treatment	Mean Weekly Fluids Consumption, mL		Mean Weekly TA Consumption, g		Mean Weekly Feed Consumption, g		Total Intake of Pb, mg/kg BW		Total Intake of Cd, mg/kg BW	
	1–6 Weeks	7–12 Weeks	1–6 Weeks	7–12 Weeks	1–6 Weeks	7–12 Weeks	1–6 Weeks	7–12 Weeks	1–6 Weeks	7–12 Weeks
Experiment 1										
Control	125.7 ^b^	158.2 ^b^	0.00 ^a^	0.00 ^a^	125.7	158.2	163.2 ^b^	141.0 ^b^	22.85	19.74 ^ab^
0.5% TA	128.3 ^b^	162.2 ^b^	0.64 ^b^	0.81 ^b^	128.3	162.2	157.1 ^ab^	143.4 ^b^	22.00	20.07 ^b^
1% TA	124.8 ^b^	163.7 ^b^	1.25 ^c^	1.64 ^c^	124.8	163.7	152.2 ^ab^	149.2 ^b^	21.31	20.89 ^b^
1.5% TA	131.0 ^b^	160.5 ^b^	1.97 ^d^	2.41 ^d^	131.0	160.5	153.6 ^ab^	145.7 ^b^	21.51	20.40 ^b^
2% TA	132.8 ^b^	167.5 ^b^	2.66 ^f^	3.35 ^f^	132.8	167.5	152.7 ^ab^	145.7 ^b^	21.38	20.42 ^b^
2.5% TA	98.50 ^a^	122.2 ^a^	2.46 ^e^	3.06 ^e^	98.50	122.2	150.0 ^a^	126.2 ^a^	21.00	17.66 ^a^
Experiment 2										
Control	147.9 ^ab^	167.5 ^b^	0.00 ^a^	0.00 ^a^	141.0 ^a^	157.4 ^c^	200.7 ^a^	164.0 ^b^	23.17 ^ab^	19.28 ^ab^
BT	136.7 ^a^	152.1 ^a^	2.74 ^b^	3.04 ^b^	148.0 ^ab^	150.2 ^b^	221.5 ^b^	153.0 ^a^	25.78 ^b^	17.85 ^a^
GT	139.5 ^a^	158.4 ^ab^	2.79 ^b^	3.17 ^bc^	147.0 ^ab^	151.8 ^b^	228.5 ^b^	163.3 ^b^	26.57 ^b^	20.20 ^b^
RT	155.4 ^b^	164.8 ^b^	3.11 ^c^	3.30 ^c^	145.0 ^ab^	140.3 ^a^	207.2 ^a^	170.7 ^bc^	22.74 ^a^	17.60 ^a^
WT	151.9 ^b^	163.4 ^b^	3.04 ^c^	3.27 ^c^	153.0 ^b^	146.7 ^ab^	214.3 ^ab^	172.3 ^c^	25.08 ^ab^	18.00 ^a^

TA—tannic acid; BT—black tea; GT—green tea; RT—red tea; WT—white tea; BW—body weight; ^a, b, c, d, e, f^ means different superscripts in the same rows differ significantly at *p* < 0.05 by Duncan’s test.

**Table 3 animals-12-02838-t003:** Mean body weight of rats at 6 and 12 weeks of exposure (mean ± SD).

	Body Weight, g		Body Weight Gain Throughout the Period of Experience
	at 6 Week	at 12 Week	
Experiment 1			
Control	231.9 ^b^ ± 22.1	336.6 ^b^ ± 17.9	105.6 ^d^ ± 3.74
0.5% TA	245.7 ^b^ ± 9.98	339.3 ^b^ ± 31.5	94.37 ^c^ ± 3.05
1% TA	246.0 ^b^ ± 15.4	329.1 ^b^ ± 25.7	83.19 ^b^ ± 1.98
1.5% TA	255.8 ^b^ ± 21.1	330.5 ^b^ ± 21.3	74.74 ^a^ ± 5.22
2% TA	261.9 ^b^ ± 17.6	344.8 ^b^ ± 9.64	83.81 ^b^ ± 4.07
2.5% TA	197.5 ^a^ ± 8.39	290.5 ^a^ ± 15.6	93.50 ^c^ ± 2.95
Experiment 2			
Control	255.6 ^b^ ± 9.05	342.6 ^b^ ± 8.35	132.0 ^bc^ ± 7.22
BT	241.1 ^ab^ ± 17.1	343.6 ^b^ ± 20.2	138.0 ^c^ ± 18.3
GT	232.4 ^a^ ± 8.23	316.4 ^a^ ± 19.1	105.5 ^a^ ± 13.7
RT	261.8 ^b^ ± 21.2	334.2 ^b^ ± 16.2	126.1 ^b^ ± 18.9
WT	256.2 ^b^ ± 20.4	343.5 ^b^ ± 30.4	133.6 ^bc^ ± 25.4

^a, b, c, d^ significant differences between groups (*p* < 0.05); SD—standard deviation.

**Table 4 animals-12-02838-t004:** Experiment 1—Relative weight of lungs, heart and brain for 6 and 12 weeks of exposure (mean ± SD).

	Relative Weight,%	
	1–6 Weeks	7–12 Weeks
Lungs		
0% TA	0.736 ^a^ ± 0.04	0.714 ^a^ ± 0.06
0.5% TA	0.707 ^a^ ± 0.06	0.715 ^a^ ± 0.02
1% TA	0.729 ^a^ ± 0.01	0.740 ^a^ ± 0.01
1.5% TA	0.713 ^a^ ± 0.02	0.737 ^a^ ± 0.02
2% TA	0.706 ^a^ ± 0.03	0.715 ^a^ ± 0.05
2.5% TA	0.933 ^b^ ± 0.08	0.838 ^b^ ± 0.03
Heart		
0% TA	0.445 ^a^ ± 0.03	0.364 ^a^ ± 0.03
0.5% TA	0.459 ^a^ ± 0.02	0.362 ^a^ ± 0.03
1% TA	0.460 ^a^ ± 0.04	0.370 ^a^ ± 0.01
1.5% TA	0.438 ^a^ ± 0.01	0.369 ^a^ ± 0.02
2% TA	0.429 ^a^ ± 0.01	0.355 ^a^ ± 0.01
2.5% TA	0.569 ^b^ ± 0.02	0.420 ^b^ ± 0.01
Brain		
0% TA	0.730 ^a^ ± 0.05	0.594 ^a^ ± 0.03
0.5% TA	0.693 ^a^ ± 0.01	0.619 ^a^ ± 0.05
1% TA	0.694 ^a^ ± 0.05	0.627 ^a^ ± 0.01
1.5% TA	0.667 ^a^ ± 0.02	0.628 ^a^ ± 0.01
2% TA	0.653 ^a^ ± 0.02	0.607 ^a^ ± 0.05
2.5% TA	0.868 ^b^ ± 0.06	0.711 ^b^ ± 0.02

^a, b^ significant differences between groups (*p* < 0.05); SD—standard deviation.

**Table 5 animals-12-02838-t005:** Experiment 1—The degree of reduction in absorbing Pb and Cd in the lungs, heart and brain (for 6 and 12 weeks of exposure) compared with the 0% TA group,%. The value of 0% TA was assumed as 100%.

	Pb		Cd	
	1–6 Weeks	7–12 Weeks	1–6 Weeks	7–12 Weeks
Lungs				
0.5% TA	−1.37 ^a^	−1.83 ^a^	−1.13 ^a^	−0.06 ^a^
1% TA	−2.26 ^b^	−2.21 ^b^	−4.91 ^b^	−6.24 ^b^ *‡
1.5% TA	−3.23 ^c^	−3.73 ^c^ ‡	−5.53 ^c^ *	−7.91 ^c^ *‡
2% TA	−4.08 ^d^	−4.18 ^d^	−15.9 ^d^ *	−22.6 ^d^ *‡
2.5% TA	−4.43 ^d^	−4.63 ^e^	−25.7 ^e^ *	−31.6 ^e^ *‡
Heart				
0.5% TA	0.00 ^a^	0.00 ^a^	0.00 ^a^	−3.53 ^a^
1% TA	−2.16 ^b^	−2.93 ^b^	−3.40 ^b^	−3.94 ^a^
1.5% TA	−3.18 ^c^	−4.48 ^c^ ‡	−5.79 ^c^ *	−6.34 ^b^ *‡
2% TA	−4.26 ^d^	−4.82 ^cd^ ‡	−12.0 ^d^ *	−19.4 ^c^ *‡
2.5% TA	−4.97 ^e^	−5.18 ^d^ *	−17.7 ^e^ *	−23.8 ^d^ *‡
Brain				
0.5% TA	0.00 ^a^	0.00 ^a^	−1.48 ^a^	−1.73 ^a^
1% TA	−0.64 ^b^	−0.75 ^b^	−4.14 ^b^	−4.22 ^b^
1.5% TA	−2.72 ^c^	−3.32 ^c^	−5.55 ^c^ *	−7.26 ^c^ *‡
2% TA	−3.35 ^d^	−5.38 ^d^ *‡	−19.2 ^d^ *	−23.3 ^d^ *‡
2.5% TA	−4.92 ^e^	−5.89 ^e^ *‡	−21.7 ^d^ *	−21.9 ^d^ *‡

^a, b, c, d, e^ significant differences between groups (*p* < 0.05); * significant versus Control groups (*p* < 0.05); ‡ 6 weeks versus 12 weeks of exposure (*p* < 0.05).

**Table 6 animals-12-02838-t006:** Experiment 2—Relative weight of lungs, heart and brain for 6 and 12 weeks of exposure (mean ± SD).

	Relative Weight,%	
	1–6 Weeks	7–12 Weeks
Lungs		
Control	0.670 ± 0.02	0.683 ± 0.01
BT	0.682 ± 0.05	0.645 ± 0.03
GT	0.659 ± 0.01	0.672 ± 0.03
RT	0.662 ± 0.01	0.657 ± 0.05
WT	0.672 ± 0.04	0.675 ± 0.02
Heart		
Control	0.435 ± 0.02	0.351 ± 0.03
BT	0.455 ± 0.02	0.338 ± 0.08
GT	0.432 ± 0.08	0.330 ± 0.01
RT	0.456 ± 0.01	0.340 ± 0.01
WT	0.441 ± 0.05	0.345± 0.02
Brain		
Control	0.670 ± 0.05	0.543 ± 0.02
BT	0.688 ± 0.09	0.541 ± 0.01
GT	0.692 ± 0.01	0.563 ± 0.05
RT	0.672 ± 0.01	0.552± 0.02
WT	0.668 ± 0.02	0.544 ± 0.04

SD—standard deviation.

**Table 7 animals-12-02838-t007:** Experiment 2—The degree of reduction in absorbing Pb and Cd in the lungs, heart and brain (for 6 and 12 weeks of exposure) compared with the Control group,%. The value of Control group was assumed as 100%.

	Pb		Cd	
	1–6 Weeks	7–12 Weeks	1–6 Weeks	7–12 Weeks
Lungs				
BT	−1.96 ^a^	−3.32 ^a^ ‡	−15.0 ^a^ *	−26.4 ^a^ *‡
GT	−2.42 ^b^	−4.18 ^b^ ‡	−16.8 ^b^ *	−27.6 ^b^ *‡
RT	−2.62 ^b^	−4.10 ^b^ ‡	−16.6 ^b^ *	−26.3 ^a^ *‡
WT	−3.68 ^c^	−4.22 ^b^ ‡	−17.4 ^c^ *	−28.8 ^c^ *‡
Heart				
BT	−4.35 ^b^	−4.78 ^b^ ‡	−15.2 ^a^ *	−17.1 ^a^ *‡
GT	−4.52 ^bc^	−4.83 ^b^	−18.4 ^c^ *	−19.6 ^b^ *‡
RT	−4.03 ^a^	−3.91 ^a^ ‡	−16.1 ^b^ *	−17.5 ^a^ *‡
WT	−4.66 ^c^	−5.01 ^b^ *‡	−19.7 ^d^ *	−25.8 ^c^ *‡
Brain				
BT	−4.38 ^a^	−4.64 ^a^	−20.4 ^a^ *	−21.7 ^a^ *‡
GT	−4.41 ^a^	−4.97 ^a^ ‡	−21.2 ^b^ *	−23.2 ^c^ *‡
RT	−4.43 ^a^	−4.75 ^a^	−20.9 ^a^ *	−22.5 ^b^ *‡
WT	−4.66 ^b^	−5.19 ^b^ *‡	−21.7 ^b^ *	−23.9 ^d^ *‡

^a, b, c, d^ significant differences between groups (*p* < 0.05); * significant versus Control groups (*p* < 0.05); ‡ 6 weeks versus 12 weeks of exposure (*p* < 0.05).

## Data Availability

The data that support the findings of this study are available from the corresponding author (A.W.-M.) upon reasonable request.

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
