# Peer review of "Tannic Acid and Tea Prevents the Accumulation of Lead and Cadmium in the Lungs, Heart and Brain of Adolescent Male Wistar Rats—Possible Therapeutic Option"

_animals, 2022, doi:10.3390/ani12202838_

Round 1

Reviewer 1 Report

Major

TA is assumed one of the crucial components of tea determining the preventive effect of tea in relation to Pb and Cd. I was confused by the results that 0.5 % TA solution demonstrated no effect on tissue Pb and Cd accumulation, while all the tested tea infusions significantly decreased tissue Cd accumulation. The TA contents in the tea infusions were far lower than those of the pure TA solutions, from 770 to 1130 mg/L, vs. 5000 mg/L.

TA content of TA in the 4 tea infusions were from 77 to 113 mg per 100 mL, while the highest concentration of the pure TA solution was 2 %, namely 20000 mg/L. For a direct comparison of TA contents between the tea infusions and the pure TA solutions, the unit should be unified as mg/L.

The Figures 2 to 5 should be improved. Name of the vertical axis should be displayed next to the charts. Bar chart with overlaid data points should be adopted to enhance transparency of the data. Sample number should be included in the legends.

Minor

Age of the rats used to the experiments should be introduced.

Line 32-33, in my opinion, the expressions including green, black, red and white tea and lungs, heart and brain are not keywords.

Line 40, what does EFSA mean?

Line 113, the effects of exposure to…

Line 144, were administered

Line 143-147, long sentence, consider revision

Line 156-157, there should be a range of TA concentration for tea infusion.

Line 175, activities of the antioxidant enzymes were determined

Author Response

Reviewer 1

TA is assumed one of the crucial components of tea determining the preventive effect of tea in relation to Pb and Cd. I was confused by the results that 0.5 % TA solution demonstrated no effect on tissue Pb and Cd accumulation, while all the tested tea infusions significantly decreased tissue Cd accumulation. The TA contents in the tea infusions were far lower than those of the pure TA solutions, from 770 to 1130 mg/L, vs. 5000 mg/L.

AU: It can be assumed that other tea components, not only TA, are responsible for the results obtained as well. Probably, the cumulative effect of various antioxidants contained in tea is the most important factor; hence, tea has greater effectiveness than the isolated TA; this information has been added in the text.

TA content of TA in the 4 tea infusions were from 77 to 113 mg per 100 mL, while the highest concentration of the pure TA solution was 2 %, namely 20000 mg/L. For a direct comparison of TA contents between the tea infusions and the pure TA solutions, the unit should be unified as mg/L.

AU: This information is included in the chapter 4.2. Antioxidant and protective effect of tea and tea components on Cd and Pb poisoning organs. I wrote „For this reason, it can be assumed that the better results in our own study were obtained using tea solutions rather than isolated TA. Although the concentration of TA in the experimental solutions was much higher than in the teas: 5000 mg/L in the 0.5%TA group - 25000 mg/L in the 2.5%TA group compared to 770 - 1130 mg/L in the groups receiving tea infusions.”

The Figures 2 to 5 should be improved. Name of the vertical axis should be displayed next to the charts. Bar chart with overlaid data points should be adopted to enhance transparency of the data. Sample number should be included in the legends.

AU: Figures corrected as suggested by the Reviewer

Minor

Age of the rats used to the experiments should be introduced.

AU: It was revised as suggested (line 27 and 29)

Line 32-33, in my opinion, the expressions including green, black, red and white tea and lungs, heart and brain are not keywords.

AU: Corrected as suggested by the Reviewer: „tannic acid; tea; antioxidant activity; lead; cadmium; rats organs; adolescent rats”

Line 40, what does EFSA mean?

AU: The abbreviation was explained. I wrote „European Food Safety Authority (EFSA)” (line 42-43)

Line 113, the effects of exposure to…

AU: It was revised as suggested.

Line 144, were administered

AU: It was revised as suggested.

Line 143-147, long sentence, consider revision

AU: It was revised as suggested. I wrote „Adolescent male Wistar rats (aged 6 weeks, weighing 210.6 ± 12.1 g) were administered black, green, red, and white tea. Cd and Pb levels (after 6 and 12 weeks of exposure) as well as SOD and CAT activity in the lungs, hearts, and brains during the combined exposure were examined in comparison with the Control group.” (line 129-133)

Line 156-157, there should be a range of TA concentration for tea infusion.

AU: It was revised as suggested. Added SD values for the determination of TA levels in different types of tea infusions

Line 175, activities of the antioxidant enzymes were determined

AU: It was revised as suggested. I wrote „Determination of antioxidant enzymes activity”

Let me thank you for your valuable comments concerning my paper

Anna Winiarska-Mieczan

Reviewer 2 Report

My comments are:

Line 19 & 20: The statement is confusing. Please elaborate and rephrase.

Line 29-31: Please check the statement for grammatical error.

Introduction Section: Introduction contains irrelevant statements and is poorly organised. Please reorganize and avoid irrelevant statements.

Line 59-64: Please elaborate the relevance of these lines with this study.

Line 68-70: Please give a brief account of the effects produced.

Line 86-87: Please either delete the words 'such as' or 'for example'.

Line 90-92: Please provide a context of these lines.

Discussion: This section is weak. Please improve this section.

Conclusions: Conclusions should not be the repetition of results. This section should be improved particularly in view of the specific findings.

Lines 576-577: Please indicate specific results of this study that lead the authors to draw the conclusion “The obtained results suggest that drinking white tea could be an ef-fective method of reducing the adverse effect of environmental pollution on the human body.”

Lines 582-584: Please name the components. Also clarify how the separate effects of those components were studied?

Author Response

Reviewer 2

Line 19 & 20: The statement is confusing. Please elaborate and rephrase.

AU: Incorrect sentence removed. The following sentence has been added „Since the tea infusions acted more effectively than the isolated TA, it can be assumed that the protective effect of teas on the organism against toxic metals should be considered in the context of the cumulative effect of various active substances present in the infusions.”

Line 29-31: Please check the statement for grammatical error.

AU: It was revised as suggested. The sentence was verified

Introduction Section: Introduction contains irrelevant statements and is poorly organised. Please reorganize and avoid irrelevant statements.

Line 59-64: Please elaborate the relevance of these lines with this study.

Line 68-70: Please give a brief account of the effects produced.

Line 86-87: Please either delete the words 'such as' or 'for example'.

Line 90-92: Please provide a context of these lines.

AU: As suggested by the Reviewer, the introduction has been rewritten. Unclear sentences removed.

Discussion: This section is weak. Please improve this section.

AU: It was revised as suggested. The discussion was modified: chapters 4.1, 4.2 and 4.3 were removed - it was considered that these subsections are not relevant to the topic and purpose of the work, and therefore unnecessarily increase the volume of the manuscript

Conclusions: Conclusions should not be the repetition of results. This section should be improved particularly in view of the specific findings.

AU: It was revised as suggested. The conclusions were modified

Lines 576-577: Please indicate specific results of this study that lead the authors to draw the conclusion “The obtained results suggest that drinking white tea could be an effective method of reducing the adverse effect of environmental pollution on the human body.”

AU: Sentence removed

Lines 582-584: Please name the components. Also clarify how the separate effects of those components were studied?

AU: These compounds have not yet been investigated by our team but we are planning to conduct such studies in a rat model. Currently, we have only analyzed TA, but we intend to conduct analogous studies of the effectiveness of catechins (predominant in tea) and quercetin.

Let me thank you for your valuable comments concerning my paper

Anna Winiarska-Mieczan

Reviewer 3 Report

The present work has aimed to determine the level of tannic acid which most effectively reduce the adverse impact of Pb and Cd on the organs of adolescent rats during combined exposure. The work, even if is not entirely new subject and some used methodology is not state-of-the-art, is well organized, complex and the results are of interest for researchers in the field. There are only minor things that must be addressed: minor English revision, references must follow the same pattern (there are some minor problems e.g. some miss the bold year...).  The ethical approval is from 2010? there is another more recent?

Author Response

Reviewer 3

The present work has aimed to determine the level of tannic acid which most effectively reduce the adverse impact of Pb and Cd on the organs of adolescent rats during combined exposure. The work, even if is not entirely new subject and some used methodology is not state-of-the-art, is well organized, complex and the results are of interest for researchers in the field. There are only minor things that must be addressed: minor English revision, references must follow the same pattern (there are some minor problems e.g. some miss the bold year...).  The ethical approval is from 2010? there is another more recent?

AU: The formatting of the References has been improved and the linguistic proofreading has been carried out. The consent of the Ethics Committee is dated in 2010, as the experiment was planned in 2010. The study was conducted in 2019, but the consent was granted for an indefinite period.

Let me thank you for your valuable comments concerning my paper

Anna Winiarska-Mieczan

Round 2

Reviewer 2 Report

The introduction, discussion and conclusion sections are not well organised as already pointed out. Discussion section looks like an introduction instead of discussing the findings of the research in context of the previous studies. Conclusions should not be a mere replication of results. It should be centered on the specific findings of the study. 

In addition, the authors did not provide any justification for the difference in age and weight of the animals used for the experiment 1 (aged 5 weeks, weighing 169.3 ± 14.7 g) and experiment 2  (aged 6 weeks, weighing 210.6 ± 12.1 g).

Author Response

Reviewer 1:

The introduction, discussion and conclusion sections are not well organised as already pointed out. Discussion section looks like an introduction instead of discussing the findings of the research in context of the previous studies. Conclusions should not be a mere replication of results. It should be centered on the specific findings of the study.

AU: The Introduction, Discussion and Conclusion chapters have been revised. Revised portion of the manuscript are marked in yellow.

In addition, the authors did not provide any justification for the difference in age and weight of the animals used for the experiment 1 (aged 5 weeks, weighing 169.3 ± 14.7 g) and experiment 2  (aged 6 weeks, weighing 210.6 ± 12.1 g).

AU:The experiments were not conducted in parallel, and experiment 1 ended first. Unfortunately, only 6-week-old rats were available when experiment 2 started. These are still growing, immature rats. I wrote „It should be taken into account that experiment 1 and experiment 2 in the present study were not conducted at the same time although both rats and the experimental environment were similar in both experiments. However, since the effectiveness of TA solutions and teas was assessed based on the degree of reduction (%) of the Cd and Pb content in the organs compared to the control group (receiving distilled water to drink), where the value was assumed to be 100%, the results of the effectiveness in both experiments were determined as comparable.” (p. 14)

Let me thank you for your valuable comments concerning my paper

Anna Winiarska-Mieczan

Round 3

Reviewer 2 Report

.